# Quantum Gravity If Non-Locality Is Fundamental

**DOI:** 10.3390/e24040554

**Published:** 2022-04-15

**Authors:** Stuart A. Kauffman

**Affiliations:** Institute for Systems Biology, University of Pennsylvania, Seattle, WA 98101, USA; stukauffman@gmail.com

**Keywords:** quantum gravity, non-locality, General Relativity, decoherence, re-coherence, actualization, von Neumann entropy, tensor networks, metric in Hilbert space, “res potentia and res extensa”, now, a quantum arrow of time, a quantum creation of spacetime, remember, record, Causal Set Theory, faithful Lorentzian manifolds, past and future light cones, UV cutoff, a phase transition from continuous to discontinuous spacetime, gamma ray spectrum discontinuities, Dark Energy, Dark Matter, singularities, Casimir effect

## Abstract

I take non-locality to be the Michelson–Morley experiment of the early 21st century, assume its universal validity, and try to derive its consequences. Spacetime, with its locality, cannot be fundamental, but must somehow be emergent from entangled coherent quantum variables and their behaviors. There are, then, two immediate consequences: (i). if we start with non-locality, we need not explain non-locality. We must instead explain an emergence of locality and spacetime. (ii). There can be no emergence of spacetime without matter. These propositions flatly contradict General Relativity, which is foundationally local, can be formulated without matter, and in which there is no “emergence” of spacetime. If these be true, then quantum gravity cannot be a minor alteration of General Relativity but must demand its deep reformulation. This will almost inevitably lead to: matter not only curves spacetime, but “creates” spacetime. We will see independent grounds for the assertion that matter both curves and creates spacetime that may invite a new union of quantum gravity and General Relativity. This quantum creation of spacetime consists of: (i) fully non-local entangled coherent quantum variables. (ii) The onset of locality via decoherence. (iii) A metric in Hilbert space among entangled quantum variables by the sub-additive von Neumann entropy between pairs of variables. (iv) Mapping from metric distances in Hilbert space to metric distances in classical spacetime by episodic actualization events. (v) Discrete spacetime is the relations among these discrete actualization events. (vi) “Now” is the shared moment of actualization of one among the entangled variables when the amplitudes of the remaining entangled variables change instantaneously. (vii) The discrete, successive, episodic, irreversible actualization events constitute a quantum arrow of time. (viii) The arrow of time history of these events is recorded in the very structure of the spacetime constructed. (ix) Actual Time is a succession of two or more actual events. The theory inevitably yields a UV cutoff of a new type. The cutoff is a phase transition between continuous spacetime before the transition and discontinuous spacetime beyond the phase transition. This quantum creation of spacetime modifies General Relativity and may account for Dark Energy, Dark Matter, and the possible elimination of the singularities of General Relativity. Relations to Causal Set Theory, faithful Lorentzian manifolds, and past and future light cones joined at “Actual Now” are discussed. Possible observational and experimental tests based on: (i). the existence of Sub- Planckian photons, (ii). *knee* and *ankle* discontinuities in the high-energy gamma ray spectrum, and (iii). possible experiments to detect a creation of spacetime in the Casimir system are discussed. A quantum actualization enhancement of repulsive Casimir effect would be anti-gravitational and of possible practical use. The ideas and concepts discussed here are not yet a theory, but at most the start of a framework that may be useful.

## 1. Introduction

Strenuous efforts to unite General Relativity and Quantum Mechanics have been underway for decades with, as yet, no clear success. Among the approaches without a background spacetime are Loop Quantum Gravity [1] and Causal Sets [2].

This article builds on the seminal papers by Cortês and Smolin, “Energetic Causal Sets” [3] and “The Universe as a Process of Unique Events” [4]. These authors propose that time is fundamental and irreversible, hence the laws are not symmetric in time. They take the “present instant” to be a primitive. The activity of time yields the next moment, causally generating a causal sequence of moments with intrinsic energy—momentum aspects. A discrete relational spacetime is to emerge from, reflect, and embed these Causal Sets.

Cortês and Smolin do not build upon quantum entanglement.

I briefly mention the work of Brian Swingle and his colleagues [5]. Theorems have established a duality between Anti-deSitter Space and Conformal Field Theories. Anti-deSitter Space does not describe the geometry of our universe. However, the theorems demonstrate that a quantum conformal field theory defined on the boundary of an Anti-deSitter Space has a dual which is a gravity in the bulk. Theory here utilizes tensor networks. Such networks encode the entanglement relations among the quantum variables. The hope is to find a conformal field theory whose dual is an appropriate spacetime.

Sean Carroll and his colleagues [6] have worked on Bulk Entanglement Gravity. Carroll and colleagues start with a set of N-entangled quantum variables in some specific pattern of entanglement. The set N is broken into two subsets. One then defines the “quantum mutual information” between the two subsets as a measure of an “area” between the two subsets. Here, the “area” is defined in Hilbert space, not physical space. Carroll and colleagues then make use of a theorem concerning a stable quantum mutual information. Then, they use a defined, classical transform, the Radon transform, to map the mutual information distances in Hilbert space to a real physical spacetime. Finally, they show a mapping to the linearized Einstein Field equations of General Relativity. In this approach, fluctuations in the “stationary” mutual information correspond to curvature in spacetime.

I now sketch the conceptual outlines of an approach that is different and based on sets of entangled quantum variables: The Quantum Creation of Spacetime from Non-Locality.

## 2. Preliminary Remarks

In General Relativity, time is a dimension. In Quantum Mechanics, time flows. One central issue is how to bridge this difference. I propose a means to do so below.Einstein remained concerned with “Now”. For Newton, “Now” is a moving point on a one-dimensional line. In General Relativity, where time is a dimension, “Now” vanishes. General Relativity has replaced Newton. “Now” has vanished from classical physics.What is “Now”? Can “Now” be real? This is a major issue. See L. Smolin’s Time Reborn [7]. I propose a solution below.General Relativity concerns only Actuals that obey Aristotle’s law of the Excluded Middle or Law of Non-Contradiction. Quantum Mechanics has features, superpositions, that do not obey the law of the Excluded Middle or Law of Non-Contradiction. By contrast, the results of actualization obey both laws. One interpretation of Quantum Mechanics is real res potentia, ontologically real possibles, and res extensa, ontologically real Actuals linked by actualization [8,9]. This is Heisenberg’s 1958 interpretation of Quantum Mechanics, where the quantum states are “potentia” [10]. “Potentia” are neither true nor false [8,9]. It is of fundamental importance that General Relativity cannot be interpreted in terms of “potentia”. All variables of General Relativity are true or false. Thus, in addition to the linearity of QM and the non-linearity of GR, the two theories may be *ontologically* distinct [8]. *Any attempt at uniting the non-Aristotelian variables in Quantum Mechanics in Hilbert space and the Aristotelian variables in General Relativity must show how this ontological distinction can be reconciled*. By taking quantum non-locality as fundamental, I propose a concrete way to do so below.General Relativity, famously, has not only black holes, but singularities within black holes where the theory simply fails. The current theory of the Big Bang must deal with an “initial singularity” just where General Relativity fails. One hope is that a theory of quantum gravity might address this. It is familiar to hope that on the Planck scale, spacetime is discrete and might obviate singularities. Can a theory of quantum gravity find other ways to eliminate singularities in addition to and perhaps beyond discrete spacetime at a smallest scale? I propose new testable ideas about eradicating singularities below.Must a theory of quantum gravity “constitute” General Relativity? Might General Relativity be emergent from but not reducible to QM? *I propose below that non-Aristotelian quantum variables create, by successive quantum actualization events, the Aristotelian spacetime among these events in which General Relativity operates.*

## 3. Starting with Non-Locality

Spatial non-locality is now firmly established and loophole free [11,12,13,14]. I take this stunning fact to be the Michelson–Morley experiment of the early 21st century. Given the Michelson–Morley experiment, Einstein (who may not have known the results) said, “Forget the Ether. Postulate that the speed of light is constant in every uniform reference frame and derive the consequences.” This move yielded Special Relativity [15].

Spatial non-locality suggests that spacetime is not fundamental [16]. Less convincingly, the fact that Quantum Mechanics can be interpreted in terms of ontologically real res potentia and ontologically real res extensa linked by actualization also suggests that spacetime is not fundamental. Potentia surely do not “seem” to be in spacetime [8,9,17]. The concept of ontologically real potentia helps conceive of something “real” that is not in spacetime [8,17].

In approaching quantum gravity, we can start with locality. Both General Relativity and String theory do with a background spacetime having locality. Loop Quantum Gravity does away with a background spacetime and seeks to build spacetime from “atoms of spacetime” [18]. Yet LQG is “local”, for it quantizes General Relativity, which is fully local, in empty spacetime. Locality in LQG shows up in “adjacent” atoms of space. Each atom is local with respect to the other [18]. Swingle and colleagues, proceeding without a background spacetime, start with the locality of a Conformal Field Theory defined on a boundary [5]. Bulk Entanglement Gravity, also without a background spacetime, similarly starts with a notion of locality, an “area” between two sets of entangled variables, here and there [6].

If we start with locality, we must explain non-locality.

Alternatively, if we start with non-locality, we need not explain it. However, we must then explain “locality”. I hope to do so. Additionally, I propose to do so without invoking a background spacetime.

I begin with the Standard Model of Particle Physics and its constants [19], perhaps evolved [7,8,20]. I also postulate that “time” is real. Then, unactualized quantum processes happen in continuous time, with no arrow of time. This is just the continuous time of Quantum Mechanics.

The aim is to use standard Quantum Mechanics. We may consider: changeable patterns of entanglement among N variables [21], decoherence [22], recoherence [23], the Quantum Zeno Effect [24] and “actualization” [25].

Begin with a set of N variables entangled in some pattern that, for now, is fixed. This can be represented by some tensor network [5]. In addition, the N-entangled variables are fully coherent. Thus, this system is entirely non-local.

## 4. The Central Postulate: Decoherence Decreases Non-Locality, So Increases Locality

Starting with N-entangled and fully coherent quantum variables, the obvious hypothesis is that decoherence decreases non-locality. However, this is an increase in “locality”. An increase in locality is a beginning of a quantum creation of spacetime, here still in Hilbert space.

The proposal that decoherence decreases non-locality is somewhat confirmed. Three papers have been published in the last few years [26,27,28]. The first paper shows that decoherence decreases non-locality as seen as violations of Bell’s inequalities. The second shows that decoherence leads to loss of entanglement and loss of non-locality. The third shows that “reconstructing spacetime” is not possible if the N-entangled variables are coherent, but it becomes possible as they decohere. I will take these articles as tentative supporting evidence for the fundamental postulate.

Recoherence is now well demonstrated [23]. N-entangled variables can undergo both decoherence and recoherence. Further, patterns of entanglement can change. The Quantum Zeno Effect (QZE) is real. Decoherence and the QZE interact. The QZE, for example, can protect from decoherence. The QZE depends upon actualization [24]. “Actualization” of quantum variables is real on most interpretations of Quantum Mechanics. It is the spot on the silver halide film in the two-slit experiment.

## 5. Deriving a Distance Metric in Hilbert Space via von Neumann Entropy

von Neumann entropy, VNE, is a measure of the extent of decoherence among N-entangled variables. VNE ranges from 0.0 continuously to ln N, where N is the dimensionality, or the number of quantum variables, in the Hilbert space. VNE is 0.0 if the N-entangled variables are fully coherent. As decoherence increases, VNE increases to ln N [29].

VNE is sub-additive: consider three quantum systems, A, B. and C. The VNE of [AB] can equal the VNE of [AC] can equal the VNE of [BC] but the sum of any two of these pairs must be less than or equal to the VNE of the set [ABC]. This sub-additivity is the triangle inequality. The sum of the lengths of any two sides of a triangle must be less than or equal to the sum of the lengths of the three sides [29]. The triangles in Hilbert space are also a *norm*. Distances between all pairs of entangled variables range from 0 to ln N. This remains true of any triangle in Hilbert space between entangled variables.


*Therefore, the VNE among a set of N-entangled variables yields a metric in Hilbert space.*


Consider any four entangled variables, A, B, C, and D, in a fixed pattern of entanglement. Let them decohere and recohere over real valued “time”. At any continuously varying instant, measure the VNE between all pairs. Any pair yields a specific VNE continuous Hilbert space distance between 0.0 and ln N. Thus, any three yield a triangle. Any four entangled variables yield a tetrahedron.

Due to use of VNE as a distance metric in Hilbert space, as decoherence increases, VNE also increases. Thus, increasing decoherence increases locality in Hilbert space.

As decoherence and recoherence occur in continuous real time among the unactualized and entangled N quantum variables, these VNE distances alter.

Changing patterns of entanglement can change von Neumann entropies. For example, entangling two variables, A and B, can change the VNE of each alone. More broadly, entangling two sets of entangled variables between the two sets can alter the total VNE. This will change the VNE distances among the N variables as will further decoherence and recoherence.

We can calculate these dynamically changing von Neumann entropies among the N-entangled variables in a tensor network [5,30]. The von Neumann entropies depend on both the specific pattern of entanglement and decoherence-recoherence.

It is important to stress that tensor networks study multiparticle quantum systems as they change entanglement patterns and decohere. There is as yet no notion of “space”. However, if we utilize the von Neumann entropy between all pairs of entangled variables as these change over time, *we have “induced” a meaning of “relative spatial locations in Hilbert space”* as these change in time. The number of particles can be finite.

I propose next that *spacetime emerges from such finite multiparticle systems*.

This proposal differs from standard quantum field theories that assign scalar or vector values to each point in a background continuous manifold. Fields are infinite dimensional.

## 6. The Emergence of Discrete Classical, i.e., Actual, Space-Time by Episodic Actualizations

The von Neumann entropy metric among N-entangled variables as they decohere, recohere, and alter patterns of entanglement is a metric in Hilbert space.

All the processes above are among unactualized quantum variables in Hilbert space. These quantum variables do not obey Aristotle’s law of the Excluded Middle or Law of Non-Contradiction. The variables are not Aristotelian. What is needed is some “transform” that maps these into an *actual classical spacetime metric* among now Aristotelian variables.

I wish to explore theory options making use of *successive quantum actualizations* to Actual events that do obey Aristotle’s law of the Excluded Middle or Law of Non-Contradiction. I wish to define a transform that yields a spacetime metric among these Actual events, where the events are “Boolean variables”, true or false. Here, spacetime will be fully relational, the relations among actual events. *The spacetime will be fully classical*. Additionally, I wish to define some transform that yields spacetime distances that reflect the von Neumann entropy distances among the still unactualized quantum variables as each actualizes.

I therefore introduce the notion of *remember*. Consider a fixed set of entangled variables: A, B, C, …, Z. At some “instant”, let one quantum variable, say A, actualize to become event “A”. At that instant, A is no longer entangled with the unactualized set B, C, …. Let this set of still unactualized variables, B, C, …, Z, with which A “was” entangled before actualization *remember* their von Neumann entropy distances to A just prior to its actualization as event “A”.

At some successive moment, let B actualize to event “B”. “B” *now converts its remembered VNE distance to “A” into a spacetime distance, say on a Planck scale, or some other scale. This creates a real spacetime distance between event “A” and event “B”.*

Let successive unactualized variables, C, D, E, … actualize. These remembered VNE distances, now converted to spacetime distances, create triangles, tetrahedra and higher ordered sets with metric distances between the remembered pairs of actualized events: [event “A”—event “B”], [event “A”—event “C”], [event “B”—event “C”] ….

As decoherence increases, VNE increases. When mapped to successive Actual events, relational spacetime distances also increase. This is consistent with the proposal that decoherence increases locality.

This process is creating an Actual or Classical discrete relational spacetime among a succession of discrete actualization events. *Here, there is a single growing actual discrete classical event spacetime*, not a quantum superposition of spacetimes as in other theories [31].

It is fundamental that each new single actual event, “X”, is added only to those previous discrete events with which it was entangled within its *remember interval*. As the discrete relational spacetime grows, and if it is generically true that not all variables are entangled, and entanglement patterns change, this will create a partial ordering among the totality of actualized events. Thus, this is a quantum stochastic birth process deeply akin to Sorkin’s Causal Set Theory [32]. In the Causal Set Theory, events are partially ordered by transitive birth order, and are causal. In the present theory, events are partially ordered by transitive birth order, but the process of becoming is not causal, it is quantum actualization. X becomes event “X”.

More, on Heisenberg’s real res potentia and res extensa linked by actualization, this “becoming” cannot be deductive: The “X is possible” of real res potentia does not entail the “X is actual” res extensa [8].

The new *remember* property can be “tuned” to endure over A actualization events, where A can be large or small. If A is large, each event will be a member of many tetrahedra. If A is small, each event will be a member of fewer tetrahedra. If A is 0, each event is isolated, connected to no other event. Here, spacetime does not exist.

A next step is to create a numerical ensemble of this construction process in order to study the set of distances among N events, as a function of A, where the total number of events and distances keep growing. As we mathematically tune A from small to large, does the set form an approximately smooth 4D spacetime manifold? Can one use multidimensional scaling or other means to find the best embedding dimension [33,34]?

## 7. A Simpler Version of *Remember*: No Adjustable Parameters

The theory can be formulated without a new adjustable parameter, A. As seen in the Quantum Zeno Effect [24], the actualized variable becomes quantum again, its amplitude flowering quadratically in time. Therefore, the simplest version of *remember* is that each quantum variable remembers its von Neumann entropy distance to all other variables with which it becomes entangled until that quantum variable itself next actualizes.

This version of *remember* has no new parameters at all beyond SU(3) × SU(2) × U(1) other than the postulate of a mapping of von Neumann distances proportionally to spacetime distances among actualized events. In this sense, the present theory is minimal.

Because spacetime distances between any pair of actualized events range from 0 to (ln N × scale length), and because VNE is sub-additive, it obeys the triangle inequality—all triangles among three events whose variables were formally entangled is a *norm*.

In a forthcoming paper [35], S. Patra and I propose a possible mechanism for *remember*. We explore the testable idea that the classical world emerges as a symmetry breaking among an initial set of 2^N^ bases to “choose one basis” by collective actualizations among N-entangled variables. An emerging basis shared among the entangled variables can also decay “slowly”. A shared presence of non-zero amplitude for a specific basis among N-entangled variables could be a basis for “remember” and vanish for each variable when it actualizes and is no longer entangled.

## 8. Relation to Causal Set Theory

The theory I propose appears to be a realization of Causal Set Theory [2,34], and to solve its fundamental problem of “faithfulness”.

The axioms of Causal Set Theory are: a set C with an order relation < is a causal set if it is: i. acyclic, ii. transitive, and iii. locally finite.

The successive actualization events in the present theory constitute an order relation, <, are acyclic, and are transitive. In addition, (iii). given a finite remember, the theory is also locally finite. The cardinality is precisely the finite number, N, of entangled variables.

Of central interest to Causal Set Theory is whether the causal set can be mapped “faithfully” to a manifold. This requires that the volume of constructed spacetime be proportional to the cardinality of the set. Remarkably, the present theory achieves this.

The metric in Hilbert space is the von Neumann entropy among the N-entangled variables, and scales as ln N. The von Neumann entropy, ln N, will be realized as the dimensionality, N, of the constructed spacetime.

The reason is striking: the process described progressively creates overlapping “complete graphs” among successions of N-entangled variables and actualized events. Each event has a distance to all the other events whose VNE distances were remembered by it. This process forms a complete graph among these events. The dimensionality of complete graphs of N vertices is “N − 1”, the simplex in which it can be embedded [35].

This is exemplified in the case of a complete unit graph on N vertices, the dimension of the system is “N − 1” and each edge is of length 1 [36]. In the present case, distances in any of the N directions will scale roughly as ln N. The dimensionality is N − 1.

In short, over successive events, the process I propose is able to form a succession of overlapping manifolds that are each “faithful”: distances are proportional to ln N, and the volume of the constructed spacetime is proportional to the cardinality, N, of the set.

This suggests that to achieve a growing four-dimensional spacetime manifold, remember should extend over five actualization events. This creates five events forming two tetrahedra with a common face and five vertices. The dimension will therefore be a spacetime of 4D. Over time, the process can add single new events thus grow successive tetrahedra on each free face of an existing set of tetrahedrons. The maximum distance in each direction should scale as ln 5, or, 1.609, the minimum distance along any direction is 0.

Specifically, consider five quantum variables, ‘A’, ‘B’, ‘C’, ‘D’, and ‘E’. Let each be entangled with all the others, forming a complete graph. Let ‘A’ actualize to “A”. Almost immediately, “A” will again become a quantum variable; call it ‘A1′. Now let ‘A1′ entangle with ‘B’, ‘C’, ‘D’ and ‘E’ *before* any of these actualize. Again, a complete graph is formed with a new member, ‘A1′. When ‘A1′ actualizes, it will participate in the further creation of a manifold with a new event, ‘A1′. *Spacetime grows by a single added event*.

In principle, this process can progressively construct self-consistent faithful manifolds with an increasing number of events, added one at a time, hence a spacetime.

Once such a spacetime has nucleated and as it grows, its statistical capacity to consistently self-propagate its own construction requires further analysis.

## 9. Now

For Newton, “Now” is a moving point on the real line that is the current determined Actual, after past determined Actuals and before future determined Actuals. In General Relativity, time is a dimension in a determined universe. General Relativity has replaced Newton. “Now” has vanished [7].

Our problem is that we persistently seek “Now” among Actuals. Cortês and Smolin do so, for example, by positing that time and the laws are irreversible among Actual events [3,4].

We need not do so.

At the instant A actualizes to event “A”, the wave function among the remaining entangled unactualized N − 1 variables changes. This is standard QM. *Let this instant be “Now”.* “Now” occurs and is shared only among the N − 1 remaining entangled variables at the instant an actualization occurs. In general, if it is true that not all variables are entangled at any time in this process, and patterns of entanglement change, this will create a partial ordering with different shared “Now” moments.

At “Now”, the amplitudes among the remaining N −1 entangled variables changes. Therefore, among the still entangled N − 1 variables their respective pairwise VNE also change.

“Now” is the frontier between “What has already non-deterministically become Actual” and “What is next non deterministically possible”. “Now” is the moment of indeterminate becoming when a Possible becomes Actual. However, this seems deeply right as we humans experience Now and time. Dowker makes the same point [37] as do Verde and Smolin [38]. The proposal here that we experience “Now”, requires that we experience actualization events, for example as qualia [39].

“Now” is the frontier between what has become Actual and what is next Possible.

A finite *remember,* however, will yield a “Thick Now” [40]. *Remember* for any variable lasts a finite time interval. During that time interval that variable may enter new entanglements with new-for-it variables and old entanglements vanish upon actualization of other entangled variables. These changing patterns of entanglement during each variable’s *remember* creates a “Thick Now” comprised of all the other now actualized variables with which it has a defined classical distance. With a Thick Now, Graph Theory becomes particularly relevant. In Erdös-Rényi random undirected graphs, a percolating giant component spans the system when each variable is randomly connected to 1.0 or more other variables [41]. During each Thick Now, if quantum variables, while they *remember*, can rapidly dis-entangle and entangle in many new, more-or-less random ways, then a persistently percolating web of the entanglement patterns present during each Thick Now, could lead to a widespread “*Simultaneous Now*” moment widely shared among all entangled variables of which one actualizes.

Each *Actual Now is comprised exactly of the set of two or more events* whose variables were entangled and actualized during the shared Thick Now of these two or more events. Any Actual Event can belong to successive overlapping sets of Actual Now.

## 10. A New Account of Non-Locality

A Thick Now, perhaps remarkably, can possibly underwrite spatial non-locality in an entirely unexpected way. The standard way to think of non-locality is within an already existing spacetime. The two entangled variables are space-like separated in this prior existing spacetime and, upon measurement, are non-locally correlated according to Bell’s inequalities.

In the present theory, spacetime is always being constructed everywhere within the growing spacetime that has already come to exist. In the present theory, unactualized variables are not in spacetime. As each is actualized, the event comes to exist in the growing relational spacetime among the events in the universe even as the constructed classical spacetime is four dimensional where time is a dimension.

However, this could possibly account for non-locality in a new way: within its finite *remember* interval, any entangled quantum variable can alter its entanglement patterns during the succession of overlapping Thick Now moments. When that variable is actualized, that new event will localize in that growing spacetime region with which that that quantum variable was most recently most and richly entangled. *Thus, two entangled variables, not in spacetime, can localize to two different forming spacetime regions that are space-like separated.* The entangled variables “construct” themselves into different ever—newly forming regions of spacetime.

In short, the von Neumann entropy distances among entangled quantum variables in Hilbert space map each variable, via successive actualization events, to its most recent entanglement history in the construction of spacetime. The quantum vacuum is ontologically real as potentia, its variables are not in spacetime, yet coordinate with the growing classical spacetime in their local construction of that spacetime.

The implications of this proposal for non-locality with respect to a role of faithful manifolds forming spacetime must eventually be evaluated in a quantitative theory, if such a theory can be constructed.

## 11. Spacetime Is Lorentzian

Importantly, via fluctuating percolating giant components, the successive addition of single new events to the partially ordered sets should be roughly Poisson within and across the partially ordered sets. Consequently, the resulting spacetime will be Lorentzian [2,42]. Kastner and Kauffman, based on Sorkin [2,42], showed that in these circumstances, Lambda should be small [9].

## 12. Views and Leibnitz’ Identity of Indiscernibles

Smolin [43] and Cortês and Smolin introduce the concept of views. In their Energetic Causal Set model [3], each discrete event has some set of past causal events. The “view” of an event is this past set, considered as a graph. Their idea is that events with identical views are identical events and cannot both exist. This realizes Leibnitz’ Identity of Indiscernibles. Their model proposes that similar views map to similar locations and minimize an energy measure thereby maximizing the difference between graphs to create a spacetime with maximally unique views.

The present theory affords a natural implementation of the “view” concept. The “view” of an event is precisely the set of quantum variables it *remembered* that themselves actualized until it actualized. These are not classically causal, but quantum correlated because upon each actualization event, the amplitudes of all remaining unactualized variables alters. Additionally, this actualization history is precisely *recorded* in the now actual relational spacetime distances among the actual events. If each event remembered and therefore is recorded by many events, and because VNE is continuous, all events will have unique views within the recorded spacetime structure.

Variables with almost the same history of entanglement will construct spacetime such that they are near one another. This instantiates the hopes of Cortês and Smolin [3] that similar event histories are reflected in how these come to be embedded in spacetime.

## 13. A Quantum Actualization Arrow of Time Recorded in the Spacetime Constructed

The Actualization process is irreversible, so the process above is irreversible. Given N fixed entangled variables merely undergoing decoherence and recoherence, there is no “record” of what happened among the quantum variables.

Magically, *there is a record in the specific relational spacetime persistently created by the processes.* There is an irreversible ongoing quantum creation of classical spacetime among the Actual Events.

On this theory, every actualization event among entangled quantum variables constructs a spacetime event that become recorded in the very relational structure of the constructed spacetime.

Thus, the very structure of the forming classical spacetime is a record of the sequential irreversible history of actualization events.

*Moreover, because the succession of actualization events is each not reversible, this sequential construction of spacetime is also a quantum arrow of time*. Therefore, the sequentially constructed discrete relational spacetime is a record of the very quantum arrow of time.

What I will call *Actual Time is the discrete succession of two or more actual events.*


*However, this is an arrow of time while the underlying laws are time symmetric. Additionally, it is an arrow of time independent of an increase of classical Entropy in the Second Law.*


Verde and Smolin [40], discuss somewhat similar ideas. They take as primitive the “in- definite becoming definite”. Hence, there is an inbuilt direction to the time. “Indefinite” can be interpreted as ontological or epistemological. This process yields a succession of transitions, indefinite to definite, which constitutes an arrow of time. By shared causal structures, a shared Now can arise.

## 14. A Role for Records

A major issue in Causal Set Theory is whether a given causal set can form a faithful manifold. It would be of interest if the following two conjectured theorems were found to hold:

*Conjecture 1:* Any causal structure unable to form a manifold does not participate in the ongoing construction of spacetime, thus leaves no record, and simply decays and vanishes. If correct, we need not be concerned about Causal Sets that do not form faithful manifolds.

*Conjecture 2*: Spacetime manifolds of different dimensions cannot mutually construct themselves. Were this true, once a four-dimensional spacetime is constructing itself, it cannot change dimension.

I remark that if forming spacetime requires complete graphs, as proposed, the probability in a random graph of forming complete graphs of increasing cardinality, N, decreases strongly with N. Regardless of the value of remember, this should bias strongly toward low-dimensional spacetimes. It seems of interest to establish possible conditions in which 4D spacetimes generically grow fastest.

## 15. Past and Future Light Cones

The existence of a set of entangled variables with a shared “Now”, according to Causal Set Theory [34], creates a *frontier between a past light cone and a future light cone of each “Actual Now” set.* The longest path between two events in the past or future light cone is “time–like”. Space-like separated sets can also be defined. This are major aspects in Causal Set Theory’s Quantum Gravity’s approach to General Relativity [34].

## 16. Percolation Theory and an Inevitable UV Cutoff

A standard expectation or aspiration for any theory of quantum gravity is a UV cutoff at some very small length scale, typically the Planck length scale of 10−35 m. Below that length scale, spacetime would not exist or would not be defined. Such a minimum length is called a UV cutoff. The presence of a UV cutoff with a minimum finite length, hence wavelength, assures a maximum finite energy scale.

A standard approach to a UV cutoff involves the metric itself, which is therefore discrete. An example is Loop Quantum Gravity which is able to define a spectrum of finite values greater than 0 for the areas and volumes of its “atoms of spacetime” [1].

The theory I present here does not have this property. Because von Neumann entropy is a *continuous variable*, the von Neumann entropy distances in Hilbert space among entangled variables are continuous, ranging from 0 to ln N. This yields triangles, tetrahedra and more complex structures in Hilbert space that are “atoms of spacetime”. *The theory maps these continuous distances in Hilbert space to continuous distances between actualized events in now classical spacetime*. *Thus, there is no minimum distance greater than*
*0 in this theory.*

Can such a theory with no minimum distance in classical spacetime have a UV cutoff? The perhaps surprising answer is “Yes, inevitably”. The fundamental reason arises from percolation theory, and percolation thresholds [44]. Percolation thresholds describe the formation of long-range connections in random systems. Below the threshold, a “giant component” does not exist. Above the threshold, a giant component exists that scales with system size [44]. On a very large two-dimensional floor tiled with white tiles, as the fraction of these are converted to black tiles, a threshold fraction, 0.59, is reached at which a connected black tile structure spans the floor. This faction is the percolation threshold. The spanning structure is the giant component [44].

In general, one considers site or bond percolation. As in the floor tile example, where the tiles are the “sites”, in the present case, site percolation is relevant. The sites are the events. There is a probability, P, that a site is, at random, occupied. For discrete systems such as the tile floor example with uniform size and shaped tiles, one considers Bernoulli random trials. For continuous systems, one considers Poisson trails [44]. The present theory adds spacetime events one at a time and creates continuous distances, hence in a Poisson process, forming Lorentzian manifolds [34].

A critical percolation threshold, *Pc*, separates subcritical systems where no giant component forms and supra-critical systems where giant components do form. Near Pc, just below it, power law distributions of many smaller and fewer larger connected clusters are typical [44]. Further below Pc, the sizes of connected clusters become smaller [44].

The fundamental point is that in any dimension and for any graph of sites, the *percolation threshold is a finite value greater than*
*0 and less than or equal to 1.0, thus 0 < Pc <= 1.0.*

The present theory constructs volumes with continuous values on any edge ranging from 0 to ln N, hence any volume must range from 0 to something like ln N D, where D is the dimension of the spacetime. *It follows that there must be volumes approaching*
*0, whose probabilities of formation, P, are smaller than any finite percolation threshold, Pc in that D dimensional spacetime. Call that volume whose probability of formation is Pc, Vc. Therefore, there is a finite volume scale, Vc, below which no giant component can form and scale with the size of the system*.

Vc specifies a minimal volume of spacetime that can percolate. Vc has the dimension, D, of the entire constructed spacetime. Were any of the edge lengths to be 0, the volume of that element would be 0 so could not percolate. Regardless of the distribution of edge lengths on any “atom of spacetime” such that its V < Vc, no percolation can occur. Therefore, there is a minimum continuous length scale in all dimensions less than or equal to D such that percolation cannot occur. *This is the UV cutoff. I stress that this UV cutoff is in continuous classical spacetime.*

The existence of a UV cutoff is inevitable in this theory. However, it is of a new type: Three issues are important.

First, *the existence of a minimal volume, Vc, that can percolate implies a phase transition in the very structure of continuous classical spacetime*. For volumes above Vc, spacetime is continuous. For volumes below Vc, each centered on a single actual event, spacetime is in discrete patches that are power law distributed for each volume size, and the power law patch sizes will decrease with smaller volumes. *It becomes of deep interest if signatures of such a phase transition can be found*. The electromagnetic spectrum is continuous. A critical volume, Vc, separates longer wave lengths and hence lower energies above Vc, and shorter wavelengths hence higher energies below Vc. Very tentative evidence for such a phase transition with respect to the *knee* in the high-energy gamma ray spectrum is described below.

Second, the present theory defines a UV cutoff in terms of a percolation threshold, Vc. but the theory also allows volumes smaller than Vc, ranging down to 0 volume. *Because there is no finite cutoff for volume size, photon wavelengths far shorter than that of the mean linear dimension of Vc are permitted. Evidence of Sub-Planckian photons noted below seems to support this.*

Third, it is often hoped that a UV cutoff will preclude the singularities of General Relativity. Loop Quantum Gravity with a discrete metric is an example [2]. It remains to be established in the present theory if this can hold. It is at least conceivable that as matter density increases, Vc can approach 0 and singularities can arise. Were this true, this hope to eliminate the singularities of General Relativity would not work. I suggest an alternative hope below.

## 17. An Emergent Metric and Ricci Tensor

In the present theory, spacetime is constructed as a sequence of actualization events with real Boolean true distances between each pair obtained by mapping von Neumann distances in Hilbert space to actual spacetime distances on some length scale. While these distances in actual spacetime are defined, this is not yet a *metric* for the spacetime. A metric, however, can easily be defined.

Propose that spacetime becomes a set of four-dimensional tetrahedra, each edge of which is 0 to ln N in length. Thus, *there is no unique shape to the atoms of spacetime*.Because actualization events occur at random moments, each possible length of each edge is equiprobable. *Thus, the distribution of volumes, lengths, and areas is uniform and stationary*.We can define the mean and variance and standard deviation of the 4D volumes in this spacetime, so also the lengths, areas and 3D volumes.We can define a Geodesic:
Preliminary definition: A “straight line” between two events is the shortest distance between the two events, 1 and 2, passing from 1 to 2 by a succession of steps along paths from 1 to 2 via near neighboring events. The shortest path from 1 to 2 is a “straight line”. The length of that path is the sum of the 4D distances between each successive step along the path from 1 to 2.A construction to show that a shortest path between two events exists that is defined as the shortest path between each pair of events along the multistep path from event 1 to event 2:
Start at event 1 and encapsulate it and several other neighboring events in a 4D Euclidian sphere of radius r. Choose r large enough such that each sphere centered on one event encapsulates several other events. Color the sphere blue. (2) For each event in the first blue sphere, again create a 4D Euclidian sphere centered on the first neighbor event and a few others close to it. Color all these second-generation spheres blue. (3) Iterate for N generations of blues spheres, each generation, N, further from event 1 than the preceding N − 1 set of blue spheres.Similarly create a set of successive pink spheres, M, around event 2.At some point, some one or more of the blue spheres will contain some events that are also in pink spheres.At that value of N and M, one or more connected paths between event 1 and event 2 exist along the blue then pink spheres.Consider the set of non-identical pathways between event 1 and event 2. Each pathway has some length defined by the mapping from von Neumann entropy distances to real spacetime distances. Among this set of “quasiminimal path- ways” from event 1 to event 2, one is the shortest between each adjacent pair of events between event 1 and event 2. Call this shortest pathway from event 1 to event 2 the “geodesic between event 1 and event two”. The length of the geodesic is given as the sum of the minimal distances between each adjacent pair of events between event 1 and event 2.An alternative is to shrink the radius, r, of all the blue and pink spheres connecting events 1 and 2 and stop shrinking r just *before* the pathway from event 1 to 2 becomes disconnected. Define the geodesic from event 1 to event 2 as the shortest total pathway from 1 to 2.Note that given a percolation threshold, Vc, as the radius, r shrinks, the set of spheres, each centered on a single event, becomes disconnected. Local connected patches of spheres at that small radius, r, are present. Further, shrinking r, corresponds to decreasing the maximum wavelength that can fit into the sphere.Hence, shrinking *r* corresponds to *increasing the minimum energy* of a quantum variable, say photon, that can fit into the sphere.*For volumes above Vc, using geodesics, define triangles in 2D subspaces* of this 4D space. As the lengths of each side, L, of the triangle increase relative to l, the variance, and the standard deviation of these lengths, L, falls off as the square root of L/l. For triangles with 10,000 l per side, L, the standard deviation in lengths is very small.*The sum of the interior angles of each triangle is increasingly well defined as L/l increases. This sum is less than 180 degrees, 180 degrees or greater than 180 degrees, thus assessing if this local region of spacetime is negatively curved, flat, or positively curved*.This is definable for 3D and 4D tetrahedra, as L/l increases. The sum of the interior angles assesses negative, flat or positive curvature.As L decreases to l, the standard deviation in all lengths increase and “angles” lose clearly defined values.
*We can now define a Ricci Tensor on an induced metric created on large L/l 4D tetrahedra.*


## 18. Hilbert Space Has a Metric Structure

In the present theory, the von Neumann entropy between all pairs among a finite set of entangled variables induces a distribution of small and large tetrahedra in Hilbert space. At each instant, there is a metric structure in Hilbert space. At each actualization event, the new event is added to the growing classical spacetime. A “Thick Now” therefore corresponds to an ongoing construction of classical spacetime with a metric that reflects the metric structure in Hilbert space during that Thick Now.

## 19. Endogenous Sources of 0 von Neumann Entropy and Possible Dark Energy

The process above among a fixed set of N-entangled variables does not stop once all are actualized. Upon actualization of A at event “A”, the variable A then becomes quantum again. This is equal to “no fact of the matter for the variable between measurements”.

However, actualization of A breaks its entanglement to the remaining quantum variables. Thus, *when A again becomes quantum, it is a pure state*. Let the now pure quantum state A entangle with some other variables.

When pure state A becomes so entangled by normal processes of the Standard Model of particle physics, the VNE of this augmented entangled set becomes lower [29]. *Thus, the newly augmented set can again decohere further and actualize yielding a further Quantum creation of spacetime.*

## 20. A Quantum Creation of Spacetime, QCS

Specifically, again, let there be only the fixed set of quantum variables A, B, … Z, actualizing and becoming quantum again among the same fixed set of variables. This is an episodic input of pure states into the system with the same set of quantum variables. The input of pure state of variable A lowers the entropy of the entangled set of variables B, C, D, and E with which A again entangles before B, C, D, and E actualize. *When A again actualizes, a single new event, “A”, is added to the growing spacetime.*

Thus, given the fixed set of quantum variables, the persistent input of pure states into the entangled set persistently lowers the entropy of the newly entangled set so enables ongoing decoherence and successive actualization events, hence the ongoing quantum creation of spacetime.

*This ongoing quantum creation of discrete actual (classical) spacetime can be an ongoing process. Such a continuous creation of classical spacetime is a candidate for Dark Energy* [45].

Why is a continuous creation of spacetime a candidate for Dark Energy? The accelerating expansion of the universe can be thought of as due to a cosmological constant which is a kind of antigravity repulsive force (negative pressure) feature of space time. Alternatively, Dark Energy can be thought of as the creation of spacetime [45].

It is very important that standard theory using zero-point energy predicts a cosmological constant 10 to 128 too large. This is said to be the worst prediction in physics. If the stochastic quantum growth of spacetime here proposed is in fact Poisson, the predicted lambda should be very small [34,42,45], as observed [45].

Two features of this theory of Dark Energy may be testable: (1) a direct experimental creation of spacetime via the Casimir effect, as discussed below; (2) if the stable input of pure state variables when they entangle with a set of variables generically lowers the von Neumann entropy of the entangled variables, and if this does not affect actualization, then the stationary volumes of spacetime that are created should be slightly biased toward smaller volumes than otherwise. Very tentative evidence with respect to the high-energy gamma ray spectrum and the *ankle* is discussed below.

## 21. Dark Matter?

Here are the properties of Dark Matter:

Dark matter is hypothesized to be: (i). non-baryonic. (ii). cold, very low velocity, (iii). dissipationless—it cannot cool by radiating photons, and (iv). collisionless—Dark Matter “particles” interact with each other only though gravity and possibly weak force [46].

We have looked for Dark Matter “particles” for decades, including axions [47] and WIMPS [48] not yet found by CERN. All such efforts assume that Dark Matter must be some form of “matter”. This assumption may be false.

The criteria for Dark Matter are consistent with the possibility that Dark Matter is spacetime itself. Therefore, a continuous quantum creation of spacetime, QCS, may be a candidate for Dark Matter.

Entirely different grounds exist to think Dark Matter may be a creation of spacetime by matter. Specifically, Chadwick et al. [49] suggest a modification of General Relativity to allow matter not only to curve spacetime but to create spacetime. For an appropriate value of their parameters, they fit Dark Matter and the excess rotation velocity in the outskirts of galaxies. This success suggests taking the hypothesis that Dark Matter really is a creation of spacetime by matter seriously.

Based on the Chadwick results [49], Kastner and Kauffman [44] suggested that the creation of spacetime by actualization could yield both Dark Matter and Dark Energy and obtained a small value for lambda. Substitute for the creation of spacetime by matter of Chadwick et al. [49] the continuous quantum creation of spacetime by decohering entangled quantum variables and actualization here suggested. Such a continuous quantum creation of classical spacetime is, therefore, a candidate for Dark Matter.

It may work. The proposed quantum creation of spacetime, QCS, is due to decoherence of entangled particles and actualization events, i.e., matter. Most of the matter in galaxies is in their penumbra, or between neighboring galaxies, where most of the Dark Matter is seen [50,51].

If a quantum creation of classical spacetime explains Dark Matter, Dark Matter need not be exotic particles. As noted, CERN has found none.

The most important issue is this: Chadwick et al. have already modified General Relativity such that matter not only curves spacetime but creates it [49]. We see below that this already hints a true formulation of quantum gravity + General Relativity modified to include a quantum creation of spacetime by matter. Further if the quantum created spacetime is Poisson, the spacetime created should be Lorentzian [34,42].

## 22. A Purely Quantum Creation of Spacetime Might Eliminate the Singularities in Black Holes

Loop Quantum Gravity proposes discrete spacetime atoms. Thus, LQG can hope to re-solve the Big Bang and Black Hole Singularities by saying that the atomic structure of spacetime prevents spacetime form shrinking to 0 volume at the singularity.

It has long been hoped that a theory of quantum gravity would somehow eliminate singularities. The paper mentioned above, [28], claims that as decoherence decreases, the reconstructed spacetime disappears into the singularity of a black hole. *This suggests that increasing decoherence, that is, the quantum creation of spacetime, might avert the disappearance of spacetime into the singularity.*

The proposal that decoherence and actualization are purely quantum creations of classical spacetime, QCS, allows us to consider that such a process inside a black hole might eradicate the singularity, where General Relativity fails. In a Black Hole, as the singularity predicted by General Relativity is approached, the density of matter increases dramatically. Thus, if the quantum creation of spacetime increases “faster” as the singularity is approached than spacetime is collapsing, this might eradicate the singularity. In short, a quantum creation of spacetime that is fast enough as the singularity is approached could eradicate the singularity. Again, if Poisson, the created spacetime would be locally Lorentzian [34,42].

## 23. A Possible Union of a Quantum Creation of Spacetime by Matter with General Relativity

The present theory is not a familiar quantum field theory assigning a value to each point in a continuous manifold. Instead, it proposes the emergence of classical spacetime from the dynamical behavior of a finite multiparticle quantum system.

Importantly, the metric structure of Aristotelian classical spacetime in this theory reflects the metric structure created by the distribution of tetrahedra in the non-Aristotelian Hilbert space of the finite number of entangled quantum particles. Processes unfold in Hilbert space, and successive actualization events construct classical spacetime with a metric structure among Actual events reflecting the metric structure in Hilbert space.

A major issue is whether such a finite dimensional Hilbert space with an induced metric can be used as a discrete approximation of a continuous quantum field theory. Alternatively, if spacetime is really discrete, no continuous quantum field theory can be correct. Perhaps our familiar field theories are approximations to some correct discrete theory.

An ultimate hope is that a finite multiparticle tensor network expressing the particle interactions of the quantum field theory, SU(3) × SU(2) × U(1), might be formulated. Were that possible, such a theory could construct the spacetime from the fundamental particles within which a modified General Relativity operates.

*Such an attempt at a theory of quantum gravity that constructs spacetime from matter is not General Relativity*. There is no construction of spacetime in General Relativity. Because General Relativity can be formulated without matter fields, General Relativity is incapable of addressing the formation of matter itself.

Matter and energy could be fully present in such a model with its quantum construction of spacetime. The different fermions form, transform, vanish, and exchange their bosons. The Higgs, so mass, is present. Spacetime is the emergent relational metric among the resulting actualization events.

It becomes an ambition to use a more mature version of the ideas discussed here to modify General Relativity, perhaps a bit like the Chadwick et al. model where matter curves and also creates spacetime [49]. On such a view, the approach to quantum gravity constructing spacetime sketched above does not constitute General Relativity, but it unites with it in a new way.

The conceptual ingredients to do so may not be too far away. The forming spacetime with fermions and bosons, and the Higgs, has matter, mass, and energy within an emerging spacetime that has a metric. Thus, there is always some stress energy tensor and a Ricci curvature tensor. It may not be too far-fetched to hope for union with General Relativity modified by a scalar field with classical probabilities derived from the amplitudes for a local construction of spacetime.


*On this emerging view, quantum gravity creates the spacetime in which General Relativity operates. In this hoped for union, “Matter tells spacetime how to grow and curve. Curved growing spacetime tells matter how to move”.*


It is critical to this theory that General Relativity emerges from but *is not reducible* to Quantum Mechanics.

It is important to note that this theory builds entirely on SU(3) × SU(2) × U(1), hence the strong and electro-weak forces of quantum field theory. There is, in this theory, as in General Relativity, no further “force”. The theory depends upon quantum actualization to construct spacetime, but “actualization” is not a force. The spacetime created by actualization of Boolean variables is not itself a quantum field theory.

## 24. Tentative Supportive Observational and Possible Experimental Evidence

I.Observational Evidence on the Granularity of Space Below the Planck Scale


*The present theory allows volumes of spacetime far below Vc, indeed down to 0. This permits photons of wavelengths far shorter than that for the dimensions of Vc. Loop Quantum Gravity, in contrast, derives a finite spectrum of atoms of spacetime with minimum area and volume.*


Wikipedia, with respect to Loop Quantum Gravity, raises the problem: “ESA’s INTEGRAL satellite measured polarization of photons of different wavelengths and was able to *place a limit in the granularity of space that is less than*
*10^–48^ m, or 13 orders of magnitude less than the Planck scale”* [52].

In Loop Quantum Gravity, spacetime is fundamentally discrete, with a minimum scale, presumably the Planck scale. As such, LQG cannot accommodate shorter wavelengths. The length scale in LQG may be tunable to 10^–48^ m, but on what grounds? Why not 10^–67^ m? *Sub-Planckian photons cast doubt on any fixed UV Energy Cutoff.*

On the present theory, spacetime and its underlying Hilbert space are both continuous. Volumes range continuously down to 0 volume. *Sub-Planckian photons are not ruled out. The observation of Sub-Planckian photons with no clear minimum length is consistent with the current theory.*

II.The High-Energy Gamma Ray Spectrum: A Statistical UV Energy Cutoff

The continuous spectrum of gamma rays is very well studied [53]. It is a featureless power law with a slope of −2.7 as energies get higher and wavelengths shorter. However, a puzzling *knee* arises at approximately 0.4 PeV whose wavelength is 10^–21^ m. For shorter wavelengths and of higher energy than the knee, the power law slope steepens to a steady −3.1 [53]. Five orders of magnitude further, at 10^18.5^ eV and a wavelength of 10^–26^ m, there is an *ankle,* where the slope increases slightly to approximately −3.0 [54].

Studies of these phenomena, intra and extra galactic, are intense. The central interests are in locating the sources for the gamma rays and accounting for their enormous energies and power law distributions.

The obvious hopes for explanations for the slope discontinuities at the knee and ankle are some kinds of discontinuities in the sources of the gamma rays such as location or composition.

However, it seems of interest to ask tentatively whether discontinuities in the very structure of spacetime might play a role.

*On the present theory, a stark discontinuity from continuous to discontinuous classical spacetime occurs at and near Vc*. As the radius, r, of spheres, each centered on a single event, shrinks beyond Vc, spacetime becomes disconnected. Might the *knee* be an abrupt alteration in slope from −2.7 to −3.1 that arises at the phase transition where classical spacetime breaks from continuous into ever smaller power law distributed patches as energies increase above 0.4 PeV?

If the *knee* does reflect a phase transition in the structure of spacetime, then the lengths of the largest tetrahedra, Vc, constructed by actualization events should be 10^–21^ m. This is far above the Planck scale and *sets the length scale* of the present theory that maps von Neumann entropy distances to real spacetime distances on a maximum length of 10^–21^ m. *Evidence that may be consistent with something special approximately* 10^−21^ m *is discussed just below.*

Five orders of magnitude smaller, at 10^–26^ m, the *ankle* arises, and the slope decreases slightly from −3.1 to approximately −3.0. It is conceivable that the *ankle* reflects the 0 Entropy input into entangled systems helping to drive Dark Energy, hence an increased abundance of very tiny volumes, so an increase in the slope from −3.1 to −3.0? If confirmed, we might have evidence for the basis of Dark Energy.

This theory is in its infancy. It is not impossible that real quantitative predictions can ultimately be made. Evidence confirming a phase transition from a continuous to a discrete structure of classical spacetime would be of the highest importance. Beginning evidence with respect to the knee is discussed below.

III.Possible Experimental Avenues via the Casimir Effect

The theory above makes a specific prediction: Actualization events among entangled quantum variables can construct spacetime. Actualization events for non-entangled quantum variables does not construct spacetime. Such actualized events are isolated.

It may be possible to test this experimentally using the Casimir effect. For parallel conducting plates, this force is always attractive [55]. Actualization of non-entangled versus entangled quantum variables between the plates constructing spacetime between the plates, so increasing the distance between the plates might be detectable as a weakening of the Casimir effect upon actualization of the entangled, but not the non-entangled variables. The Casimir effect falls off as the 4th power of the distance between the plates.

Conversely, the repulsive Casimir effect is a form of “levitation”, so antigravitational [56,57]. It may be possible to test for an enhanced repulsive Casimir effect via actualization events constructing spacetime among entangled but not non-entangled variables. Direct demonstration of antigravitational effects would be stunning and of interest to Captain Kirk.

I recently co-authored with colleagues a study of the consequences of *removing* a single-mode photon between the two parallel plates of the Casimir system as a function of the distance between the plates and the energy of the photon ranging up to the Planck energy [58]. As photon energy goes up, the distance between the plates shrinks and asymptotically *reaches a length far above the Planck length*: (L/Lp)^2/3^. Here, L is the distance between the plates, Lp is the Planck length.

For a distance, L, between the plates of 10 microns, which is common in Casimir systems, the distance between the plates shrinks 10^–20^ m. For lower energy photons, the distance will shrink approximately 10^–21^ m or less. This suggests that there may be something special approximately a length scale of 10^–21^ m.

However, 10^–21^ m is also the energy of the gamma ray *knee,* 0.4 PeV, and thus it is also the proposed length scale, far above the Planck length scale, at which a transition from continuous to discrete classical spacetime occurs. In short, if m is the length scale below which classical spacetime becomes discontinuous, classical spacetime connected patches should be power law distributed at each length scale and have smaller connected patches as length scale decreases. If we propose that photons of any wavelength can only be emitted from continuous spacetime patches large enough to hold that wavelength, the number of emitted photons should decrease as a power law in decreasing wavelengths below a phase transition to discontinuous spacetime patches. If the *knee* is the discontinuity, at 0.4 PeV, *this is now testable with the extant data confirming the observed onset at the knee of a steeper power law rate of decreasing numbers of photons observed ever further below the knee. With a better quantitative theory, this should predict the precise increase slope of the power law to −**3.1 below the knee.*

## 25. Disproof of the Theory

In some theories of quantum gravity, it is not clear what the novel predicted observables are [52]. In the present case, a quantum creation of spacetime is an essential novel predicted observable. If there is no quantum creation of spacetime by actualization events, the theory is false. If none can be found, the theory is weakened. The theory necessarily has a critical volume, Vc. If no evidence can be found for a phase transition in the structure of classical spacetime from continuous to discrete as length scales decrease the theory is weakened or false.

## 26. Future Directions

The present body of work is based on a fixed and unchanging number, N, of quantum variables. This limitation is not necessary. A manuscript in preparation and privately shared reports that the variables of the Standard Model, SU(3) × SU(2) × U(1), considered a set of transformations among the variables, is formally capable of collective autocatalysis. With mild assumptions this can lead to an exponentially autocatalytic increase in the number of quantum variables whose stochastic dynamics can break matter-antimatter symmetry to yield baryogenesis. Increasing entanglement and actualization among these variables then become a candidate for Inflation in the early universe. A natural cessation of autocatalysis and Inflation can occur as spacetime inflates. This leads to a transition in which Dark Matter then Dark Energy, discussed above, play sequential natural roles. These ideas may provide a natural transition from SU(3) × SU(2) × U(1) to the Lambda CDM model of Cosmogenesis.

Singh and Doré, in a recent article [59], suggest a similar theory in which a fixed total set of quantum degrees of freedom exist. Among these, an increasing number become entangled and drive Inflation.

## 27. Conclusions

I have taken non-locality to be the Michelson–Morley experiment of the early 21st century, assume its universal validity and try to derive its consequences. Spacetime, with its locality, cannot be fundamental, but must somehow be emergent from entangled coherent quantum variables and their behaviors. There are, then, two immediate consequences: i. if we start with non-locality, we need not explain non-locality. We must instead explain an emergence of locality and spacetime. ii. There can be no emergence of spacetime without matter. These propositions flatly contradict General Relativity, which is foundationally local, can be formulated without matter, and in which there is no “emergence” of spacetime.

I have proposed that quantum gravity cannot be a minor alteration of General Relativity, but it demands its deep reformulation: matter not only curves spacetime, but “creates” spacetime.

This quantum creation of spacetime consists of:Fully non-local entangled coherent quantum variables.The onset of locality via decoherence.A metric in Hilbert space among entangled quantum variables by the sub-additive von Neumann entropies between pairs of variables.Mapping from distances in Hilbert space to a classical spacetime by episodic actualization events.Discrete spacetime is the relations among these discrete events.Now is the shared moment of actualization of one among the entangled variables when the amplitudes of the remaining entangled variables change instantaneously.The discrete episodic irreversible actualization events constitute a quantum arrow of time.The arrow of time history of these events is recorded in the very structure of the spacetime constructed.Actual Time is a succession of two or more actual events.

These processes create a single growing spacetime similar to Sorkin’s Causal Set Theory [2,32,34] with partial ordering of these successive quantum birth-becoming events. Because the addition of single events here is Poisson, and partially ordered, the resulting spacetime should be Lorentzian [2,32,34,42]. The theory here proposed should generically and sequentially construct “faithful” manifolds. The existence of a shared “Now” implies a past and future light cone for the variables sharing each “Actual Now”. The past and future light cones each have a timeline structure [34]. Each “Now” propagates faithful manifolds, successively constructing its future light cone.

The theory inevitably yields a UV cutoff of a new type, a phase transition from percolating continuous classical spacetime above the threshold volume, Vc, to discrete classical spacetime below the threshold.

This quantum creation of spacetime may be able to modify General Relativity and may account for Dark Energy, Dark Matter, and the possible elimination of the singularities of General Relativity.

The proposed quantum creation of spacetime occurs in the presence of matter and energy, with volume and density, hence a stress energy tensor and a Ricci tensor. Therefore, some mapping to a modified General Relativity may be possible as a scalar field with a probability for a creation of a “local volume” of discrete relational spacetime. This is not unlike the theory of Chadwick et al. [49].

If in classical physics General Relativity Matter tells Spacetime how to curve, and curved Spacetime tells Matter how to move, a revised theory including quantum gravity in which matter itself constructs spacetime may be: Matter tells Spacetime how to grow and curve, and curved growing Spacetime tells Matter how to move.

Possible observational tests with respect to Sub-Planckian photons and the *knee* and *ankle* phenomena in high-energy gamma rays, and experimental tests in both the attractive and repulsive Casimir effect setting are described. A quantum actualization enhancement of repulsive Casimir would be anti-gravitational and of possible practical use.

The ideas and concepts discussed here are not yet a theory, but at most a framework that may be useful to develop a full quantitative theory.

## Data Availability

No supporting data.

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
