# Peer review of "Quantum Gravity If Non-Locality Is Fundamental"

_entropy, 2022, doi:10.3390/e24040554_

Round 1
Reviewer 1 Report
The author take non-locality to be the Michaelson Morley experiment of the early 21st Century, assume its universal validity, and try to derive its consequences.
The paper is good to be published in entropy but it must be presented differently. I suggest to write longer sentences. and to cite more people in the introduction .
Author Response
Thank you. Please see response to Reviewer 2. The author hopes that these answer your concerns.
Reviewer 2 Report
In this paper the author sets the stage for the formulation of a novel theory that can address the existing tension between General Relativity and a quantum theory. The proposal can be developed to the formulation of a quantum theory of gravity based on the causal set theory. The idea borrowed by other authors (Cortes and Smolin etc.) is interesting but the presentation presents some issues and must be improved.
The first point that should be clarified is that the tension between GR and a quantum theory is caused by the fact that GR is a classical physical theory, hence the discussion about Aristotele’s exclusion middle law can obviously not be applied in this context.
The idea that something “real” that is not spacetime is common to other theories such for instance LQG and DSR. The idea underlying these theories is that the real physics takes place in the momentum space and is relational, the spacetime emerges as an useful representation to describe physics. The references at line 166 are about pure quantum effects (in the sense of particle physics) and do not tackle gravity, it may be useful to include references about LQG and DSR.
A problem concerning LQG is about the fact that it is not yet known if this theory admits GR as a low energy limit, such for example GR admits SR in the limit of low energy (that is mass) and low curvature (line 176 – 177).
The idea of using VNE as a measure is intriguing, but the definition of a norm should be improved: a norm must be positive defined, the triangle inequality must be true and these properties are verified. But a norm must be even non degenerate, otherwise one is defining a pre-norm. The paragraph at lines 245-249 should be better explained. The distance between two points can range from 0 to log (N), hence (line 401) the distance between two points can be only finite? Perhaps it could be better to give an explicit example on how to map the von Neumann entropy to a spacetime norm.
The introduction of spacetime must be improved: it is not clear if spacetime introduced in this proposal is quantized or not. In LQG the thetraedron is quantized, but in this proposal it is not clear explained if the introduced spacetime is deterministic. The actualization of quantum potential events determines a classical spacetime, but how the quantum structure is lost in the actualization process?
The presentation of causal set theory can be improved as a review. It is not simple for a reader to understand some technical properties and terms not defined or explained. This issue is particularly evident in the definition of the concept of Thick Now and the constructed light cone with simultaneity.
The used concept borrowed from percolation theory must be introduced (Poisson and so on).
The idea that the actualization process is not reversible and this can determine a time arrow is really intriguing, but it must be better explained and introduced. The Vc minimal volume that introduces a limit between classical and quantum spacetime is interesting, but it is not well explained how percolation can introduce a transition phase phenomenon that separates classical and quantum spacetime. In this aspect perhaps only a precise mathematical formulation of the theory can clarify this point. A question: Vc seems to be energy dependent, in this sense the separation of classical and quantum spacetime can disappear if a physical system has enough energy? But in this case the GR singularities can not be avoided. Moreover the quantum gravity effects could not be detected since only high energy particles can probe the Planck regime, but the Vc decreases with high energies. Shrinking matter can increase coherence? In this case VNEà0 and Vcà0, hence the singularity in a black hole is again present. In the same aspect the UV cutoff disappears (and the renormalization problem is again present) increasing the mass in the Universe.
Using the cosmic gamma rays to detect such effects can be real problematic, since there are too many experimental uncertainties about cosmic messengers and their propagation.
A last important question: it is possible to show if the actualization process can admit GR as large limit of actualizing events number?
Considered all the issues present in the paper I suggest a major revision tackling all the critical points before publishing this interesting work.
Best Regards,
the Reviewer
Reviewer 3 Report
No comments.
Author Response
Reviewer 3 made no comments.
Round 2
Reviewer 2 Report
The second version of the paper is surely an improvement respect to the first version and some issues are addressed in a correct way.
Nevertheless, in my opinio some critical points should be again tackled and I enumerate them in the following.
Firts point - a norm must be subaddictive, homogeneous of degree one and must be equal to 0 only for the null element:
1) || x + y || ≤ || x + z || + || z + y || ∀ x, y, z ∈ V (considered space)
2) || k x || = | k | || x || ∀ k ∈ R (real numbers)
3) || x || = 0 ⇒ x = 0
these properties allows one to define a distance, that should verify the following equalities:
d(x, y) ≤ d(x, z) + d(z, y) (triangular disequality)
d(x, y) = d(y, x)
d(kx, ky) = |k| d(x, y)
d(x, y) = 0 ⇔ x=y
If only the triangular disequality is respected, one is dealing with a pre-norm. Perhaps it can be useful to stress that the definition of the norm via the von Neumann entropy can respect all the definition properties to obtain a real norm. (for instance line 383 - 385).
Second point - now it is clear that the spacetime is classical, but can be discrete since the percolation can take place only above a threshold energy. But this point is in my opinion again not enough clear with respect to the dependency on the energy of the process. I mean: the UV cut-off obtained in this way should have a probabilistic character, if the percolation does not take place the actualization of two different events belonging to the Hilbert space present a minimal distance. But this minimal distance seems to depend on the energy of the events, hence the UV cut-off is energy dependent? In this case an enough energetic probe can feel the spacetime with a minimal lenght approaching zero and therefore the cut-off diverges? (this means for high energetic probes the cut-off disappears? Therefore, the UV catastrophe is again present?)
The lackness of a precise mathematical definition of the distance obtained via the von Neumann entropy can be the cause of such issue about the paper. On this point I can suggest to the author the following paper, where a precise mathematical construction is given, even if in the context of causality in discrete spacetime:
| arXiv:gr-qc/0004026 |
I can also suggest some papers about DSR theories that can be cited in the context of theories that consider the momentum space as the real space where the Physics take place and the coordinate space is considered as emerging and not fundamental:
arXiv:1101.0931
| arXiv:1210.7834 |
arXiv:1106.5710
arXiv:1906.05595
This research is interesting and proposes some intriguing ideas. If the author can address the underlined critical points, I think this paper can be published in Entropy.
Best Regards
the Reviewer
